# Influences of Lavender Essential Oil Inhalation on Stress Responses during Short-Duration Sleep Cycles: A Pilot Study

**DOI:** 10.3390/healthcare9070909

**Published:** 2021-07-18

**Authors:** Wakako Yogi, Mana Tsukada, Yosuke Sato, Takuji Izuno, Tatsuki Inoue, Yoshiki Tsunokawa, Takayuki Okumo, Tadashi Hisamitsu, Masataka Sunagawa

**Affiliations:** 1Department of Physiology, School of Medicine, Showa University, Tokyo 142-8555, Japan; wkkyg0613@cmed.showa-u.ac.jp (W.Y.); tizunosuke19831113@gmail.com (T.I.); tatsuki22@med.showa-u.ac.jp (T.I.); t.yoshiki@med.showa-u.ac.jp (Y.T.); tokumo@med.showa-u.ac.jp (T.O.); tadashi@med.showa-u.ac.jp (T.H.); suna@med.showa-u.ac.jp (M.S.); 2Pharmaceutical Department, Showa University Hospital, Tokyo 142-8666, Japan; 3Department of Neurosurgery, School of Medicine, Showa University, Tokyo 142-8666, Japan; yosukens@med.showa-u.ac.jp; 4Department of Urology, School of Medicine, Showa University, Tokyo 142-8666, Japan

**Keywords:** α-amylase, antistress effect, aromatherapy, chromogranin A, cortisol, lavender essential oil

## Abstract

Lavender essential oil (LEO) was reported to improve sleep quality. We investigated the influence of aromatherapy by testing the effects of LEO on stress responses during a short-duration sleep in a single-blind, randomized, crossover trial. The subjects were twelve healthy adults who were nonsmokers without any known disease and who were not prescribed medications, and nine of these completed the study. After the subjects had fallen asleep, they were sprayed with LEO using an aroma diffuser. Before and after 90 min of sleep, α-amylase, chromogranin A (CgA), and cortisol levels in saliva were measured as objective stress indicators, and the Japanese version of the UWIST Mood Adjective Checklist was used as a subjective indicator. A comparison of changes before and after sleep, with and without LEO, revealed that the cortisol level did not significantly change; however, α-amylase (*p* < 0.05) and CgA (*p* < 0.01) levels significantly decreased after LEO inhalation. A mood test indicated no change in mood before and after sleep, with or without LEO. Since α-amylase and CgA reflect the sympathetic nervous system response, these results indicate that LEO aromatherapy during a short-duration sleep cycle suppresses the stress response, especially that of the sympathetic nervous system.

## 1. Introduction

Aromatherapy is an alternative therapy that is generally defined as the application of essential oils to cure symptoms associated with discomfort and illnesses. The term “aromatherapy” was coined around 1930 by French chemist René-Maurice Gattefossé after he discovered that lavender essential oil (LEO) effectively healed his burned hand [1]. Aromatherapy is currently used to treat various symptoms, such as stress relief, sleep disturbances, and anxiety [2,3,4]. However, scientific evidence to support these effects has not been completely explained. In this study, we focused on LEO. There are several varieties of lavender, but the most commonly known variety of lavender is *Lavandula angustifolia* belonging to Labiatae family. *L. angustifolia* is a short semiligneous plant that is frequently used in aromatherapy. The major components of *L. angustifolia* are linalool, linalyl acetate, geranyl acetate, myrcene, 1.8-cineole, and camphor [5]. *L. angustifolia* possesses various therapeutic properties, such as antibacterial [6], anti-inflammatory [7], analgesic [5], antistress [8,9], anxiolytic [10], and sedative [11] effects. It has also been reported to accelerate the onset of sleep and improve sleep quality [12,13]. Goel et al. [14] reported that LEO inhalation increased slow-wave (deep) sleep, and Arzi et al. [15] reported that LEO odor sensed via a nasal mask exhibited a trend of reducing wake frequency.

Sleep is an important component of recovery from various physiological conditions; however, many people suffer from chronic sleep deprivation [16,17], which affects mood, cognitive function, performance, and homeostasis [18,19]. Because of the reports of the beneficial effects of taking naps, some offices have permitted nap sessions during work shifts. Depending on the job type, such as a long-distance driver or overnight hospital staff, naps may be required during work. Napping improves work and exercise performance [20] and reduces psychological stress [21]. However, the beneficial effects of napping seem to differ depending on the length of the nap. A short-duration sleep session of 20–30 min has been reported to promote wakefulness and improve work and exercise performance [22,23], whereas a 60–90 min nap, during which the person experiences slow-wave sleep and rapid eye movement sleep, has the same learning effect as an 8 h sleep regarding memory, especially perceptual skills [24]. Conversely, it was reported that a long daytime nap (60 min or more/day) was associated with a higher risk of cardiovascular disease [25].

As mentioned above, many people suffer from chronic sleep deprivation and sleep disorders [16,17]. General treatments include medications that induce or prolong sleep; however, the drugs may not improve sleep quality and may cause side effects such as addiction or drug resistance [26]. A safe therapy with few side effects is needed; therefore, essential oils, which are considered safe and have few side effects, have promising potential as a therapeutic agent for treating sleep disorders [27]. This study aimed to clarify whether LEO utilized as aromatherapy could improve sleep quality during short-duration sleep such as a nap, with a focus on the stress response.

## 2. Materials and Methods

### 2.1. Subjects

All subjects gave their informed consent before they participated in the study. The study was conducted according to the Declaration of Helsinki, and the protocol was approved by the Ethics Research Committee of Showa University (approval number, 3183; date of issue, 27 July 2020). The subjects were 12 healthy adults between the ages of 27 and 45 (32.42 ± 6.83 years), male (*n* = 3; 41.67 ± 8.50 years) and female (*n* = 9; 29.33 ± 1.80 years), who were nonsmokers without basic disease and were not prescribed with any medication. In particular, they were previously confirmed not to have allergies to essential oils, asthma, sleep apnea and nasal diseases. We explained in advance to the subjects that they could not drink alcohol within 24 h of the experiment, and should avoid caffeine intake after waking up until the test. Additionally, breakfast had to be consumed within 1 h after waking up, and water intake was prohibited 1 h before the experiment. All subjects received verbal and written information about the study before signing an informed consent form.

### 2.2. Aromatherapy

The LEO (*Lavender angustifolia* L., Sigma-Aldrich, 61718, St. Louis, MO, USA) treatment used in the study was prepared as follows: 30 μL LEO was dissolved in 500 mL distilled water and sprayed with an aroma diffuser (wyy009, JANRI, Saitama, Japan). We confirmed that LEO was completely dissolved in the solution by the absence of the Tyndall phenomenon [28]. The diffuser was placed in the direction of the subject’s head, about 20 cm away from the bed. After detecting slow-wave patterns on an electroencephalograph (EEG), LEO was sprayed around the sleeping subjects. Only distilled water was sprayed as a control. To determine the concentration of LEO, a preliminary experiment was conducted with the cooperation of several subjects who did not participate in the main study with reference to previous reports [29,30,31]. In the preliminary experiment, we noticed that subjects would wake up if the concentration was high (>60 μL/500 mL), thus this was a factor we considered in the performance of the study.

### 2.3. Study Design

The crossover trial was conducted in 2021 (Figure 1). All subjects participated in the study three times, and they were examined on two separate occasions (aromatherapy and control tests). On the first day, a preliminary experiment without LEO inhalation was conducted to see if the subjects could adapt to the laboratory environment and achieve a sleeping state. A pure sleep waveform without LEO intervention was obtained. The main tests were conducted on the second and third days. Subjects were divided into two groups: a group that experienced intervention on the second day with no intervention on the third day and a group without intervention on the second day that experienced intervention on the third day. The subjects were randomly allocated in advance, and a single-blind study was conducted. For the allocation of the subjects, a computer-generated list prepared by Microsoft Excel 2019 (Microsoft Corporation, Redmond, WA, USA) was used. There was a 1–2 week washout period between each experiment day.

### 2.4. Condition of Sleep

The conditions of sleep were that the subject went to bed between 11 p.m. and 1 a.m. on the day before the experiment and woke up between 6 a.m. and 8 a.m.

The experiment was conducted at 10 a.m. in the laboratory of the Department of Physiology, Showa University School of Medicine (Tokyo, Japan). The room had no windows and maintained an average humidity of 25% ± 5% with an average temperature of 25 °C. The sleep state was confirmed by EEG. All subjects underwent HD-EEG with the 256-channel Geodesic Sensor Net (Electrical Geodesics, Inc., Eugene, OR, USA) and then slept for approximately 90 min.

### 2.5. Evaluation of Stress

#### 2.5.1. Objective Index: Measurements of Salivary Stress Markers

Salivary samples were collected using Saliva Collection Aid (5016.02; Salimetrics, Carlsbad, CA, USA) and Cryovial (5004.02; Salimetrics) immediately before sleeping and immediately after waking up. Subjects were instructed to expectorate into the vail any saliva that naturally secreted in their mouths. The samples were kept in the refrigerator until they were centrifuged at 4000 rpm for 10 min, and their supernatants were stored at −80°C until the analyses were conducted. For each subject, the cortisol level was analyzed using the commercial Cortisol Enzyme Immunoassay Kit (1-3002; Salimetrics). The sensitivity of the assay was <0.007 μg/dL. The intra- and inter-assay coefficients of variation (CVs) were <10%. The α-amylase level was measured using the commercial Salimetrics Salivary Alpha-Amylase Assay Kit (1-1902; Salimetrics). The assay sensitivity was <0.4 U/mL, and the intra- and inter-assay CVs were <10%. The chromogranin A (CgA) concentration was measured by the Human Chromogranin A EIA Kit (YK070; Yanaihara Institute Inc., Shizuoka, Japan). The assay sensitivity was <0.14 pmol/mL, and the intra- and inter-assay CVs were <5%. Finally, all measured values were corrected based on the protein contents in the saliva sample, because these values are affected by saliva production. Accordingly, the total protein concentrations in the saliva were determined using the Pierce™ BCA Protein Assay Kit (23227; Thermo Scientific, Waltham, MA, USA). All measurement procedures were conducted according to the manufacturer’s instructions.

#### 2.5.2. Subjective Index: The UWIST Mood Adjective Checklist

According to the basic emotion theory, the basic emotions are joy, fear, surprise, dis-gust, anger, and sadness, and as per the dimension theory, emotions are expressed as a dimensional vector such as pleasure–discomfort or arousal–sleep [32]. Based on these theories, several scales have been developed to evaluate the state of emotion. We used the Japanese version of the UWIST Mood Adjective Checklist (JUMACL) to obtain descriptions about mood [33]. The original version of the checklist was created by Matthews et al. [34], which is used in many studies [35,36,37,38,39]. By answering a small number of questions, the current mood and emotions could be easily evaluated, and minimizing the subject’s burden. In addition, this checklist was created based on the dimension theory, making it possible to assess the arousal level (energy arousal and tension arousal). The present study aimed to evaluate the changes before and after sleep; therefore, this checklist was selected to evaluate the short-term psychological state of the patient, rather than the long-term psychological state [40]. A total of 10 factors (active, bright, dull, energetic, idle, industrious, passive, sleepy, unenterprising, and vigorous) that were related to stress, were used, which are also used to evaluate energy arousal. Responses were scored using a scale from 4 corresponding to “definitely” to 1 corresponding to “definitely not”. The level of energy arousal was indicated from the total score of 10 factors [33].

### 2.6. Statistical Analysis

The age of subjects was presented as the mean ± standard deviation, and all other experimental data are presented as the median (25% and 75% percentiles). The statistical significance of the differences was evaluated via the Wilcoxon signed-rank test using the statistical software JMP Pro 15.0 (SAS Institute, Cary, NC, USA). All *p*-values of <0.05 were considered statistically significant.

## 3. Results

### 3.1. Salivary Stress Markers

The levels of salivary cortisol, α-amylase, and CgA before and after sleeping were measured and corrected based on the protein concentrations (Table 1). The levels of cortisol and α-amylase significantly decreased after 90 min of sleep, both with and without lavender inhalation. However, the CgA level increased both with and without lavender, although there were no significant differences.

Next, we compared the rates of after/before change between with and without lavender when the value before sleeping was 100%. With or without LEO, 90 min of sleep reduced the salivary cortisol level, and there was no significant difference (*p* = 0.13; Figure 2A). However, LEO inhalation significantly reduced the salivary α-amylase level (*p* < 0.05; Figure 2B). After 90 min of sleep, CgA secretion tended to increase; however, LEO inhalation significantly suppressed the increase (*p* < 0.01; Figure 2C).

### 3.2. UWIST Mood Adjective Checklists

The 10 factors affected by stress were scored with points ranging from 1 to 4, which were compared before and after sleeping, with and without lavender inhalation. Further, the levels of energy arousal expressed by the total score of 10 factors were compared in the same way. No significant change was observed for any factor and energy arousal (Table 2).

## 4. Discussion

Sleep deprivation and mental and physical stress are involved in the onset and exacerbation of various symptoms and illnesses; however, there are many people who cannot avoid these experiences in modern society. A nap may be able to solve these problems, especially if the quality of sleep is enhanced. To determine whether LEO aromatherapy could improve sleep quality, we utilized salivary cortisol, α-amylase, and CgA as objective stress markers. Unlike blood, saliva can be collected noninvasively without mental or physical stress. The hypothalamic–pituitary–adrenal (HPA) axis, which involves the endocrine system, and the sympathetic–adreno–medullar (SAM) axis, which involves the sympathetic nerve, are well-known [41,42]. Activation of the HPA axis promotes cortisol secretion, and the cortisol levels in plasma and saliva are correlated [43,44]. Salivary α-amylase is secreted by the salivary glands, especially the parotid gland, under direct stimulation by the sympathetic nerve, and noradrenaline secreted by the adrenal medulla [45,46]. CgA is mainly present in adrenal medulla chromaffin cells, which contain secretory granules of sympathetic nerve endings, and is a glycoprotein involved in the storage and secretion of catecholamines; therefore, CgA is presumed to reflect the reaction of the SAM axis [47]. Salivary CgA is produced in the submandibular gland, and CgA secretion increases as mental stress loads increase, such as performing a cognitive task [48] or speaking [49]; however, CgA does not increase with physical stress load. Therefore, CgA secretion is presumed to respond specifically to psychological stress and be less responsive to physical stress.

The stress-relieving effect following a short-duration sleep cycle could not be subjectively confirmed with the results of this study (Table 2); however, the cortisol and α-amylase levels were significantly inhibited regardless of the presence or absence of LEO inhalation (Table 1). Therefore, a nap may be able to suppress the excitement of the endocrine and sympathetic nervous systems, resulting in an antistress effect. No change was observed in the evaluation using JUMACL, because longer sleep duration would be required for the subjective perception of stress reduction. The UWIST was also used to evaluate sleep-related moods [50,51]; however, none of them evaluated changes before and after short-term sleep. Therefore, it was not suitable the evaluation in the present study. In the future, we will consider using other evaluation methods.

When the rate of change in each marker was compared with and without LEO, cortisol secretion exhibited a slight inhibitory tendency after LEO inhalation, but no significant change was identified (Figure 2A). Conversely, secretions of α-amylase and CgA, which reflect changes in the sympathetic nervous system, were significantly suppressed by LEO inhalation (Figure 2B,C). These results suggest that LEO has a greater inhibitory effect on the stress response of the sympathetic nervous system than on that of the endocrine system. Linalool, which is a major component of *L. angustifolia*, is considered a major factor in the pharmacological effects of lavender [2,52,53]. Linalool acts as a competitive antagonist of glutamate that binds to the N-methyl-D-aspartate (NMDA) receptor. Exposure to LEO enhances the affinity of GABA for the GABA_A_ receptor in the mouse brain, which in turn decreases excitability [54,55]. GABA_A_ and NMDA receptors are expressed in sympathetic neurons and in neurons of the upper central nervous system that regulate the sympathetic nervous system [56]. Therefore, linalool may act on NMDA and GABA_A_ receptors, which cause decreases in α-amylase and CgA secretions.

Of particular interest in the results of this study was that the CgA level after a short-duration sleep was higher than that before sleep, although there was no significant difference, and LEO inhalation significantly suppressed the increase (Figure 2C). Den et al. [43] reported that salivary CgA levels peaked upon awakening and then quickly decreased to the nadir after 1 h and maintained a low level throughout the day. However, the secretory dynamics that occur during sleep and the influence of napping on the secretion have not been clarified. Although the function of increased CgA secretion upon awakening is unclear, CgA secretion appeared to increase even after a short sleep. Furthermore, the effect of suppressing CgA secretion with LEO inhalation is unknown, so further investigation is needed in the future. One possibility is that aromatherapy using LEO may enhance the antistress effects of napping. As mentioned above, a long nap (60 min or more during the day) increases the risk of cardiovascular disease [25]. Recently, clinical studies have suggested that CgA may be involved in cardiovascular pathologies, including hypertension, heart failure, myocardial infarction, and acute coronary syndromes [57]. Since the relationship between plasma and salivary CgA is unclear, the influence of LEO inhalation on plasma CgA secretion should be investigated to determine whether it is possible to reduce the above-mentioned risks by administering LEO while napping.

There are some limitations to this study. First, the number of subjects was small, and there was a bias because of the age and gender ratio of the subjects. This study had another aim to establish the feasibility of a future trial. Therefore, within 6 months, we selected as many subjects as possible and conducted the experiment. We plan to increase the number of subjects and to conduct research using other essential oils in future studies. Second, we need to clarify whether the influence on the stress response differs depending on the duration of the nap. In this single-blind study, we planned to objectively determine the effect of LEO during sleep. In our preliminary experiments, the inhalation of LEO during light sleep resulted in arousal, although the concentration of LEO was lower than that used in other studies [28,29,30]. Therefore, we started LEO inhalation after confirming the appearance of slow waves. Slow waves generally appear about 30 min after falling asleep [58]. Moreover, if we wake up during slow-wave sleep, we feel drowsy and tired because of sleep inertia [59]; therefore, the average time when the first slow-wave sleep ends, i.e., 90 min, was used in the present study. In the future, after reconsidering the research methods and conditions, we would like to investigate the effect of LEO on shorter naps, too. Third, the nap conditions may have affected the results. For example, sleeping in the laboratory is different from sleeping at home. Therefore, further studies with adjustments made to these factors can overcome these limitations.

## 5. Conclusions

The application of LEO as an aromatherapy during short-duration sleep may suppress the stress response, especially that of the responses of the sympathetic nervous system, even if there is no change in the subjective symptoms. It was suggested that sleep quality can be improved by using LEO aromatherapy for short periods of sleep and when taking a nap.

## Figures and Tables

**Figure 1 healthcare-09-00909-f001:**
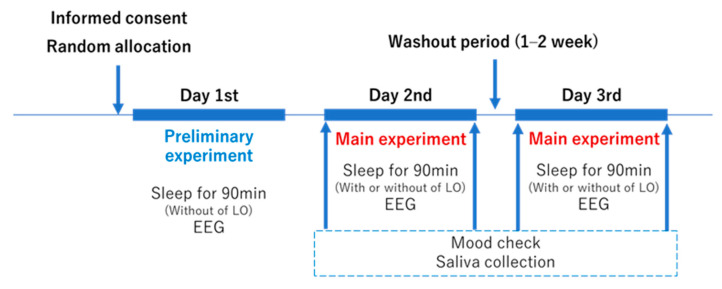
Experiment schedule. LEO, lavender essential oil; EEG, electroencephalogram.

**Figure 2 healthcare-09-00909-f002:**
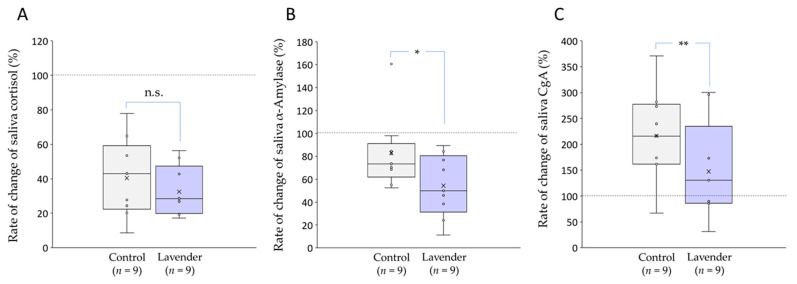
Rate of change in salivary stress markers. (**A**) Cortisol; with or without LEO, 90 min of sleep reduced the salivary cortisol levels, and there was no significant difference. (**B**) α-amylase; LEO inhalation significantly reduced the salivary α-amylase level (* *p* < 0.05). (**C**) CgA; the CgA levels after 90 min of sleep increased with and without LEO; however, LEO inhalation significantly suppressed the rate of increase (** *p* < 0.01). Horizontal lines within boxes denote median values, and x marks denote the mean values. n.s., not significant.

**Table 1 healthcare-09-00909-t001:** Changes in salivary stress markers.

(*n* = 9)		Before	After	*p*-Value
Saliva cortisol (nmol/mg protein)	Control	170.33 (31.89, 252.03)	34.55 (17.04, 55.06)	0.010 **
	LEO	70.33 (68.41, 112.52)	21.72 (19.54, 27.03)	0.000 **
Saliva α-amylase (U/mg protein)	Control	119.59 (82.56, 152.46)	80.76 (76.87, 106.69)	0.031 *
	LEO	104.64 (82.99, 126.49)	63.79 (48.55, 80.51)	0.001 **
Saliva CgA (pmol/mg protein)	Control	4.39 (3.07, 8.07)	9.61 (8.63, 14.60)	0.143
	LEO	8.32 (5.46, 10.06)	12.95 (6.91, 17.39)	0.230

All values were corrected based on the protein concentrations and are expressed as median (25% and 75% percentiles). The *p*-value was analyzed before and after 90 min of sleep. * *p* < 0.05, ** *p* < 0.01 (the Wilcoxon signed-rank test). CgA, chromogranin A; LEO, lavender essential oil.

**Table 2 healthcare-09-00909-t002:** UWIST Mood Adjective Checklists.

(n = 9)		Before	After	Rate of Change(After/Before)	Before vs. After *p*-Value	Control vs. LEO *p*-Value
Bright	Control	2 (1, 3)	1 (1, 3)	1.00 (1.00, 1.00)	1.000	0.750
	LEO	2 (1, 2)	2 (1, 2)	1.00 (1.00, 1.00)	1.000	
Vigorous	Control	2 (1, 3)	1 (1, 2)	1.00 (0.50, 1.00)	0.375	0.375
	LEO	2 (1, 2)	2 (1, 2)	1.00 (1.00, 1.00)	1.000	
Sleepy	Control	2 (2, 3)	3 (2, 4)	1.00 (1.00, 1.50)	0.375	0.375
	LEO	2 (2, 3)	2 (2, 4)	1.00 (1.00, 1.33)	1.000	
Energetic	Control	2 (2, 3)	1 (1, 3)	1.00 (0.50, 1.00)	0.500	0.250
	LEO	2 (1, 3)	2 (1, 3)	1.00 (1.00, 1.00)	1.000	
Idle	Control	4 (3, 4)	4 (3, 4)	1.00 (1.00, 1.00)	1.000	1.000
	LEO	4 (3, 4)	3 (3, 4)	1.00 (1.00, 1.00)	0.500	
Industrious	Control	2 (1, 3)	1 (1, 2)	1.00 (1.00, 1.00)	0.500	0.375
	LEO	2 (1, 3)	2 (1, 2)	1.00 (1.00, 1.00)	1.000	
Unenterprising	Control	4 (4, 4)	4 (3, 4)	1.00 (1.00, 1.00)	1.000	1.000
	LEO	4 (4, 4)	4 (3, 4)	1.00 (1.00, 1.00)	1.000	
Passive	Control	3 (3, 4)	4 (3, 4)	1.00 (1.00, 1.00)	1.000	0.500
	LEO	4 (2, 4)	4 (3, 4)	1.00 (1.00, 1.33)	0.750	
Dull	Control	2 (1, 3)	3 (2, 4)	1.00 (1.00, 1.33)	0.125	0.500
	LEO	3 (2, 3)	3 (2, 4)	1.00 (1.00, 1.50)	0.375	
Active	Control	2 (1, 3)	2 (1, 4)	1.00 (1.00, 1.33)	0.500	0.500
	LEO	2 (1, 3)	2 (1, 3)	1.00 (1.00, 1.00)	1.000	
Energy arousal	Control	25 (23, 28)	26 (25, 26)	1.00 (1.00, 1.17)	0.438	0.742
	LEO	25 (24, 29)	25 (25, 27)	1.00 (0.93, 1.07)	0.984	

The 10 factors were scored using a scale from 4 corresponding to “definitely” to 1 corresponding to “definitely not”. The levels of energy arousal were expressed by the total score of 10 factors. No significant changes were observed for any of the factor-related stress before and after sleep, with or without LEO. All values are expressed as the median (25% and 75% percentiles). LEO, lavender essential oil.

## Data Availability

Data are available from the corresponding author, upon reasonable request.

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
