# Peer review of "Influences of Lavender Essential Oil Inhalation on Stress Responses during Short-Duration Sleep Cycles: A Pilot Study"

_healthcare, 2021, doi:10.3390/healthcare9070909_

Round 1
Reviewer 1 Report
This is an interesting study which investigated the influence of aromatherapy by testing the effects of Lavender Oil on stress responses during a short-duration sleep in a single-blind, randomized, crossover trial.
However, there were some unclear points.
- It was written that that a long daytime nap (60 min or more/day) was associated with a higher risk of cardiovascular disease in Introduction part, but subjects slept for about 90 minutes in this study. Why it was for approximately 90 minutes, not for 20-30 minutes?
- UWIST Mood Adjective Checklist (JUMACL) is not well explained and it is not clear why JUMACL was used for this study. Furthermore, the result of JUMACL evaluation was not well discussed. Please cite previous studies which used JUMACL for sleep evaluation and discuss the validity of the results and methods.
- Of the 12 subjects, three females were excluded from the study because they were unable to produce a sufficient volume of saliva for the measurements.
Saliva is difficult to secrete when the sympathetic nervous system is dominant, but isn't it possible that this experiment itself caused stress to the patient? Since it is difficult to replicate the actual situation in sleep research, I think this point can be discussed more in the Limitation section as an issue for future research.
Author Response
Dear Reviewer 1:
We would like to thank you for your review.
We have revised our manuscript according to your suggestions.
Sincerely,
Point 1: It was written that that a long daytime nap (60 min or more/day) was associated with a higher risk of cardiovascular disease in Introduction part, but subjects slept for about 90 minutes in this study. Why it was for approximately 90 minutes, not for 20-30 minutes?
Response 1: Thank you for your precise opinion. As it is known, scent affects emotions, and our bodies tend to relax with our favorite scent. However, in this single-blind study, we planned to objectively investigate the effect of LO during sleep. In our preliminary experiments, the inhalation of LO during light sleep resulted in arousal, although the concentration of LO was significantly lower than that used in other studies. Therefore, in this study, we started LO inhalation after confirming the appearance of slow waves based on EEG measurements. Slow wave generally appears about 30 min after falling asleep. Moreover, if we wake up during slow-wave sleep, we feel drowsy and tired because of sleep inertia; therefore, the average time when the first slow-wave sleep ends, i.e., 90 min, was used in the present study.
However, as you pointed out, the effect of LO on 20–30 min of sleep should also be examined. Therefore, in the future, after reconsidering the research methods and conditions, we would like to investigate the effect of LO on shorter naps as well. We have added this description in the text (Line 292~).
Point 2: UWIST Mood Adjective Checklist (JUMACL) is not well explained and it is not clear why JUMACL was used for this study. Furthermore, the result of JUMACL evaluation was not well discussed. Please cite previous studies which used JUMACL for sleep evaluation and discuss the validity of the results and methods.
Response 2:
(Line 153~) We have added a description for UWIST as well as the reason for choosing it. In addition, energy arousal has also been added to the evaluation items.
(Line 249~) We have added some thoughts on the results. We will consider using other evaluation methods in the future.
Point 3: Of the 12 subjects, three females were excluded from the study because they were unable to produce a sufficient volume of saliva for the measurements.
Saliva is difficult to secrete when the sympathetic nervous system is dominant, but isn't it possible that this experiment itself caused stress to the patient? Since it is difficult to replicate the actual situation in sleep research, I think this point can be discussed more in the Limitation section as an issue for future research.
Response 3: As you mentioned, it is possible that stress reduced the amount of saliva produced; however, there are individual differences in the amount of saliva produced. In this study, naturally secreted saliva was collected, instead of stimulated saliva (such as via the gum method or Saxon method), to minimize stress on the subjects, which usually produces less amount of saliva than stimulated saliva. At least 1 mL of saliva was required to measure all stress markers used in this experiment; therefore, on the 1st day in the preliminary experiment, subjects who could not collect 1 mL of saliva were excluded. This study included the comparison of stress states before and after sleep, and the how it is affected by lavender oil; therefore, we think that the evaluation is possible using this protocol. However, as pointed out, sleeping under experimental conditions and usual sleep are different, so this result only suggests one possibility. In the future, we would like to consider an experiment wherein the subjects collect saliva at home. We have included this description in the limitations (line 302).
Reviewer 2 Report
This study describes the effect of lavender oil on some stress responses in short sleep periods. The work would be strengthened by addressing the following:
- While the language is generally acceptable there is a need to do a final check of grammar
- The Abstract refers to 12 subjects however only 9 actually completed the study
- Lines 39-43 require the addition of references to the literature
- Lines 65-69 – this text is unclear and would benefit from rewriting
- The study sample size is very small – please provide sample size calculations to demonstrate that statistically valid data can eb obtained from 9 individuals
- As essential oils do not dissolve in water how did the authors ensure that the oil was dissolved in distilled water at lines 87-88
- Please provide further explanation of the process at lines 92-95 and how concentration was determined
- Line 106 – what randomisation process was used
- The data for the study appears to be pooled data rather than based on the change for each individual. This may also explain why the SD in Table 1 is very large. I recommend the authors look at change in parameters rather than using pooled data or all before/after measures
- Table 2 – the score here is based on 1-4 scale. Why was parametric analysis used for nonparametric data especially when there are only 9 data points
- It is hard to judge the validity of the conclusions given the small sample size and pooling of data for analysis
Author Response
Dear Reviewer 2:
We would like to thank you for your review.
We have revised our manuscript according to your suggestions.
Sincerely,
Point 1: While the language is generally acceptable there is a need to do a final check of grammar.
Response 1: Thank you for your comment. The manuscript has been proofread again.
Point 2: The Abstract refers to 12 subjects however only 9 actually completed the study.
Response 2: We have corrected the text by adding the following:“and nine of these completed the study.” (Line 20).
Point 3: Lines 39-43 require the addition of references to the literature.
Response 3: References have been added [2-4].
Point 4: Lines 65-69 – this text is unclear and would benefit from rewriting.
Response 4: (Line 66) We agree with you and after our discussion, we came to the conclusion that the first sentence was difficult to understand, so we deleted it and replaced it with the following sentence:
As mentioned above, many people suffer from chronic sleep deprivation and sleep disorders.
Point 5: The study sample size is very small – please provide sample size calculations to demonstrate that statistically valid data can eb obtained from 9 individuals.
Response 5: This study has a pilot element, with an aim to establish the feasibility of a future trial. Therefore, within 6 months, we selected as many subjects as possible and conducted the experiment. Thus, statistically significant values were obtained, and the feasibility of the experiment might be established in this study. In the future, we plan to increase the number of subjects and conduct research using other essential oils. This description has been added in the limitations (line 287~).
Point 6: As essential oils do not dissolve in water how did the authors ensure that the oil was dissolved in distilled water at lines 87-88.
Response 6: A total of 30 μL of LO was added to 10 mL of water and mixed using the vortex mixer, and then this solution was diluted with 490 mL of water. We finally used LO at a concentration of 30 μL/500 mL. We visually confirmed that no separation occured before and after LO inhalation.
LO contains about 50% of the main components linalyl acetate and linalool [5]. Considering the characteristics of these main components, their densities (linalyl acetate, 0.895; linalool, 0.86) are approximately equal to that of lavender oil (0.882). The solubility at a temperature (25°C) as that in the laboratory is 30 mg/L for linalyl acetate and 1.6 g/L for linalool. In this study, 30 μL of LO was dissolved in 500 mL of water, so the specific gravity conversion was about 53 μg/L, considering that the aromatic components were sufficiently dissolved.
Point 7: Please provide further explanation of the process at lines 92-95 and how concentration was determined.
Response 7: Preliminary experiments were conducted with several subjects who did not participate in this study to determine the concentration of LO. According to previous reports [25–27], the effects of inhalation of LO have been investigated at various concentrations from 0.05% to 10%. In our preliminary experiment, when the concentration was high (>60 μL/500 mL), a few subjects awakened. Accordingly, the final concentration was decided. This description has been inserted in the text (Line 97).
Point 8: Line 106 – what randomisation process was used.
Response 8: For allocation of the subjects, a computer-generated list prepared by Microsoft Excel 2019 (Microsoft Corporation, Redmond, WA, USA) was used. Subjects were randomly allocated into two groups (LO inhalation on day 2 or 3) and assigned at a ratio of 1: 1. These details have been inserted in the text (line 110).
Point 9: The data for the study appears to be pooled data rather than based on the change for each individual. This may also explain why the SD in Table 1 is very large. I recommend the authors look at change in parameters rather than using pooled data or all before/after measures.
Response 9: Table 1 displayed pooled data; we used Wilcoxon signed-rank test for statistical analysis to evaluate changes before and after sleep. Therefore, we changed the average value to median (25% and 75% percentiles).
Point 10: Table 2 – the score here is based on 1-4 scale. Why was parametric analysis used for nonparametric data especially when there are only 9 data points.
Response 10: Similar to Table 1, Wilcoxon signed-rank test was used to compare the values before and after sleep, and the rates of changes were compared with and without lavender oil. The values have been changed as that in Table 1.
Point 11: It is hard to judge the validity of the conclusions given the small sample size and pooling of data for analysis.
Response 11: All experimental data are presented as mean ± standard deviation. All results were analyzed by Wilcoxon signed-rank test. As you stated, the small sample size is the weakest point of this research, and as mentioned in the limitation (Line 287~), we will increase the sample size in the future.
Reviewer 3 Report
Dear authors,
Please see the attached file comments and revised the manuscript point-by point.

Author Response
Dear reviewer 3,
We would like to thank you for your detailed review.
We have revised our manuscript according to your suggestions.
Point 1: (Line 1) Insert ‘Essential’
Comment 1: ‘Lavender Oil’ in the title was changed to ‘Lavender Essential Oil’.
Point 2: (Line 16) Please use the LEO as abbreviation.
Comment 2: As you pointed out, the abbreviation of lavender Oil was corrected to LEO. I also replaced all LOs in the text below with LEO.
Point 3: (Line 42) Labiatae →Labiatae family
Comment 3: We Added ‘family’.
Point 4: (Line 43) for → in
Comment 4: We changed ‘for’ into ‘in’.
Point 5: (Line 73 & 116) Please clearly mentioned all inclusion and exclusion criteria.
Please add this items in inclusion and exclusion criteria.
Comment 5: All inclusion and exclusion criteria are described in ‘Subject’ (Line 73), and only conditions related to sleep are described in ‘Condition of Sleep’ (Line 116).
Point 6: (Line 173-174) Please delete this sentence.
Comment 6: We deleted it according to your suggestion.
Point 7: Table 1
Comment 7: The notation in Table 1 has also been changed to LEO and the explanation has been added in the legend.
Point 8: Figure 2
Comment 8: The vertical labels were changed to ‘Rate’.
‘NS: not significant’ was added in the figure legend.
Point 9: Table 2
Comment 9: The notation in Table 2 has also been changed to LEO and the explanation has been added in the legend.
Point 10: Table 2 Please delete the underline of Energy arousal.
Comment 10: As you pointed out, it was removed.
Point 11: (Line 204) Please clearly mentioned the scores characteristics.
Comment 11: It was added in the legend.
Point 12: It is better change the old references with newer ones.
Comment 12: We changed the old ones that can be changed. However, we left the ones that were not suitable for others.

Round 2
Reviewer 2 Report
Thank you for your responses; I feel however that some of the responses have not addressed the comments made previously.
- Sample size – while statistical analyses scan be done on small sample sizes, and they may produce statistically significant results, this does not mean that the results are valid from the perspective of error types. The question remains as to what a priori sample size calculation was completed to ensure this was a valid analysis.
- LO and distribution in water – I find it difficult to believe that the authors could discern visually whether or not 30 uL of LO in 10 mL of water was completely dissolved, or that it did not come out of solution between mixing and when the spraying occurred. This is a limitation of the study as the oil may or may not have been dissolved in the water.
- Data analysis - the authors have not addressed my previous question of why they completed pooled analysis rather than looking at before/after change of individuals. It is also not clear how the three repeats of the study for each participant was treated in analytic terms.
- Table 1 - Given the wide distribution and that the overall change before/after for the controls seems to be of the same magnitude (or bigger) compared with the before/after change for the intervention group it is questionable whether the intervention was any better than the control. From the data presented the conclusion that LO treatment has benefits in short term sleep is not justified.
- Presentation of data – the authors have not addressed why non-parametric data have been presented as mean and SD. If the authors acknowledge in their inferential analyses that this is non-parametric data then it is not appropriate to present descriptive statistics using parametric measures.
Author Response
Dear reviewer 2,
We would like to thank you for your review.
We have revised our manuscript according to your suggestions.
Point 1: Sample size – while statistical analyses scan be done on small sample sizes, and they may produce statistically significant results, this does not mean that the results are valid from the perspective of error types. The question remains as to what a priori sample size calculation was completed to ensure this was a valid analysis.
Comment 1: Thank you for your precise opinion. The required sample size could not be calculated because there was no previous research that could be used as a reference for our study. As we commented last time, this study also aimed to establish a research method, so we collected as many samples as possible during a specific period. We plan to increase the sample size with reference to the issues you pointed out (Line 287~).
Point 2: LO and distribution in water – I find it difficult to believe that the authors could discern visually whether or not 30 uL of LO in 10 mL of water was completely dissolved, or that it did not come out of solution between mixing and when the spraying occurred. This is a limitation of the study as the oil may or may not have been dissolved in the water.
Comment 2: As mentioned last time, considering the solubility of the main components, it is considered that at least the main components are sufficiently dissolved in the 30 μL/ 500 mL solution. Just in case, the state of dissolution was confirmed by the Tyndall phenomenon. The 30 μL/ 10 mL solution seemed to be dissolved visually, but the Tyndall phenomenon was confirmed (the optical path of the laser beam could be clearly seen) (Picture left in the attached file). In other words, as you pointed out, it is not completely dissolved. However, the Tyndall phenomenon could not be confirmed in the very low concentration solution (30 μL/ 500 mL); dissolved in 500 mL of water after sufficiently suspending the 30 μL/ 10 mL solution (Picture center).
For reference, the case of water is also shown (Picture right). From the above, it is considered that LO is uniformly dissolved at a ratio of 30 μL / 500 mL. This was added in Line 90.
Point 3: Data analysis - the authors have not addressed my previous question of why they completed pooled analysis rather than looking at before/after change of individuals. It is also not clear how the three repeats of the study for each participant was treated in analytic terms.
Comment 3: The data in Table 1 were statistically analyzed between before and after sleep, and then, the rate of change before and after sleep was statistically analyzed between with and without LO (Figure 2) by a paired nonparametric test (the Wilcoxon signed-rank test). We added after / before in Line198 to make it clear that we compared the rate of change of individuals. We have confirmed with statistical experts, Dr. Hirotaka Ochiai (Department of Hygiene, Public Health and Preventive Medicine, Showa University School of Medicine), that the statistical analysis method is appropriate.
I am sorry that the latter is difficult to understand (see Figure 1). We conducted a crossover test using the same subject.(Line 103~)The first day was the trial that was conducted to help the subject get used to the laboratory and to see if the subject could sleep there. Subjects were divided into two groups: a group that experienced intervention on the second day with no intervention on the third day and a group without intervention on the second day that experienced intervention on the third day. The data on the second day and the data on the third day were compared (Figure 2).
Point 4: Table 1 - Given the wide distribution and that the overall change before/after for the controls seems to be of the same magnitude (or bigger) compared with the before/after change for the intervention group it is questionable whether the intervention was any better than the control. From the data presented the conclusion that LO treatment has benefits in short term sleep is not justified.
Comment 4: The main conclusions of the present study were based on the results in Figure 2. As mentioned above, the same subject is undergoing a crossover study, and the rate of change of each parameter before and after sleep is statistically analyzed between with and without intervention by the Wilcoxon signed-rank test. In the paired rank test, we do not think that the wide distribution of values is a problem. As a result of the above, the levels inα-amylase and CgA decreased significantly in the intervention group, and the inhalation of LO is considered to be useful for these factors. If our interpretation seems wrong, could you please tell me again?
Point 5: Presentation of data – the authors have not addressed why non-parametric data have been presented as mean and SD. If the authors acknowledge in their inferential analyses that this is non-parametric data then it is not appropriate to present descriptive statistics using parametric measures.
Comment 5: Thank you for pointing out. We made a mistake in the description of Line 176, so we corrected ‘mean’ to ‘median’.